# The Acute Impact of External Compression on Back Squat Performance in Competitive Athletes

**DOI:** 10.3390/ijerph17134674

**Published:** 2020-06-29

**Authors:** Mariola Gepfert, Michal Krzysztofik, Maciej Kostrzewa, Jakub Jarosz, Robert Trybulski, Adam Zajac, Michal Wilk

**Affiliations:** 1Institute of Sport Sciences, Jerzy Kukuczka Academy of Physical Education in Katowice, 40-065 Katowice, Poland; m.gepfert@awf.katowice.pl (M.G.); m.krzysztofik@awf.katowice.pl (M.K.); m.kostrzewa@awf.katowice.pl (M.K.); jaroszjakub88@gmail.com (J.J.); a.zajac@awf.katowice.pl (A.Z.); 2Department of Medical Sciences, The Wojciech Korfanty School of Economics, 40-065 Katowice, Poland; rtrybulski@o2.pl; 3Provita Zory Medical Center, 44-240 Zory, Poland

**Keywords:** blood flow restriction, resistance exercise, power output, bar velocity, performance, occlusion

## Abstract

The aim of the present study was to evaluate the effects of external compression with blood flow restriction on power output and bar velocity changes during the back-squat exercise (SQ). The study included 10 judo athletes (age = 28.4 ± 5.8 years; body mass = 81.3 ± 13.1 kg; SQ one-repetition maximum (1-RM) 152 ± 34 kg; training experience 10.7 ± 2.3 years). Methods: The experiment was performed following a randomized crossover design, where each participant performed three different exercise protocols: (1) control, without external compression (CONT); (2) intermittent external compression with pressure of 100% arterial occlusion pressure (AOP) (EC-100); and (3) intermittent external compression with pressure of 150% AOP (EC-150). To assess the differences between conditions, the participants performed 3 sets of 3 repetitions of the SQ at 70% 1-RM. The differences in peak power output (PP), mean power output (MP), peak bar velocity (PV), and mean bar velocity (MV) between the three conditions were examined using repeated measures two-way ANOVA. Results: The post hoc analysis for the main effect of conditions showed a significant increase in PP (*p* = 0.03), PV (*p* = 0.02), MP (*p* = 0.04), and MV (*p* = 0.03), for the EC-150, compared to the CONT. Furthermore, a statistically significant increase in PP (*p* = 0.04), PV (*p* = 0.03), MP (*p* = 0.02), and MV (*p* = 0.01) were observed for the EC-150 compared to EC-100. There were no significant changes in PP, PV, MP, and MV, between EC-100 and CONT conditions. Conclusion: The results indicate that the use of extremely high-pressure external compression (150% AOP) during high-loaded (70% 1-RM) lower limb resistance exercise elicits an acute increase in power output and bar velocity.

## 1. Introduction

External compression causing blood flow restriction is a modern training tool used during resistance exercises [1,2,3]. The amount of external compression used for blood flow restriction is adjusted based on the individual value of arterial occlusion pressure (% AOP) [4]. The value of 100% AOP is the point where blood flow is completely cut off. Furthermore, the effect of external compression on blood flow restriction (BFR) is also related to a range of individual characteristics (limb circumference, body composition) as well to the type of cuff (tourniquet shape, width, and length) [5,6,7].

Previous studies have shown that external compression can be used before exercise as ischemic preconditioning [2,8,9,10,11,12,13] or during exercise [2,14,15]. The main mechanisms responsible for the adaptive responses associated with training under compression and BFR include increased metabolic stress [16,17], which is a consequence of the compression of the vasculature proximal to the skeletal muscle, which results from an inadequate oxygen supply (hypoxia) within the muscle tissue [18,19]. Previous studies indicate that external compression causing blood flow restriction increases the effectiveness of resistance exercises, however most studies concern chronic adaptive changes [20,21,22,23,24,25,26,27,28]. There are only a few studies related to the acute impact of external compression on performance during resistance exercise [5,29,30,31,32], and only one considered the impact of BFR on power output and bar velocity changes [14]. Wilk et al. [14] indicated that intermittent, high-pressure external compression increases power output and bar velocity during the bench press exercise at 70% 1-RM. However, such a positive effect was observed only when a wide cuff was used (10 cm), and such changes were not observed with the narrow cuff (4 cm) at 90% of AOP. This suggests that the effectiveness of external compression may be mainly conditioned by the width of the cuff. 

Furthermore, as suggested by Wilk et al. [14], increased performance during exercise with external compression may also be associated with the mechanical energy generated by the cuff [33]. A cuff is a passive element, but during the eccentric phase of the movement it can produce additional elastic energy, which is returned during the concentric phase of the movement [14,33]. Therefore, not only physiological factors but also mechanical energy generated by the cuff can be a potential factor that influences the magnitude of acute changes during exercise under external compression. This statement was also confirmed by Rawska et al. [29] and Wilk et al. [32]. Rawska et al. [29] showed an increase in the number of performed repetitions (5 sets) during the bench press exercise at a load of 80% 1-RM under external compression (80% AOP), compared to control conditions. Similarly, Wilk et al. [32] showed that extremely high pressure of external compression (150% AOP) significantly increased the result of the 1-RM test and the strength-endurance performance during the bench press exercise. However, such a positive effect was observed only at a pressure of 150% AOP and not at the lower value of 100% AOP. This indicates that pressure above 100% AOP can significantly enhance exercise performance. However, as Wilk et al. [32] suggested, these results cannot translate to other types of exercises or to lower limbs, which requires further research.

One of the most common and effective exercises for lower limb hypertrophy, strength, and power development is the back squat (SQ). The SQ is often a basic exercise in resistance training programs as well as a powerlifting competition [33,34,35,36,37]. Although the SQ is considered a highly effective exercise for developing the power output of the lower limbs [35,38], no studies have assessed the impact of external compression with BFR on acute changes during the SQ. Due to the lack of scientific data concerning the acute effects of external compression on lower limb performance, the aim of the present study was to evaluate the effects of external compression with full BFR on power output and bar velocity during the SQ. Like Wilk et al. [32], to isolate the potential effect of mechanical energy generated by the cuff, we decided to use two values of pressure, both causing the same, complete shutting down of blood flow (100 and 150% AOP). It was hypothesized that an external compression would increase power output and bar velocity during the SQ.

## 2. Methods

### 2.1. Study Design

The experiment was performed following a randomized crossover, counterbalanced design, where each participant performed three different testing protocols: (1) control, without external compression (CONT); (2) intermittent external compression with pressure of 100% AOP (EC-100); and (3) intermittent external compression with pressure of 150% AOP (EC-150). Before testing, two familiarization sessions and one session dedicated to the evaluation of 1-RM were performed. The entire research procedure lasted 6 weeks with a 4–5-day interval between each trial. During the experimental sessions, the subjects performed 3 sets of 3 repetitions of the SQ at a load of 70% 1-RM. All testing sessions were performed in the Strength and Power Laboratory at the Academy of Physical Education in Katowice, Poland. 

### 2.2. Subjects

Male judo athletes (n = 10) with resistance training experience of 10.7 ± 2.26 years volunteered for the study after completing an informed consent form (age = 28 ± 5.75 years; body mass = 81 ± 13 kg; SQ 1-RM = 152 ± 34 kg; 1-RM/BM = 1.9 ± 0.4 kg/kg). The inclusion criteria were an SQ personal record of at least 150% body mass. The study participants were allowed to withdraw from the experiment at any moment and were free from musculoskeletal disorders. The subjects were instructed to maintain their normal dietary habits throughout the study and not to use any supplements or stimulants for the duration of the experiment. The athletes were informed about the benefits and potential risks of the study before providing their written informed consent for participation. The study protocol was approved by the Bioethics Committee for Scientific Research, at the Academy of Physical Education in Katowice, Poland (no. 02/2019), and performed according to the ethical standards of the Declaration of Helsinki, 1983. 

### 2.3. Procedures

#### 2.3.1. Familiarization Session and the 1-RM Strength Test

Three weeks before the main experiment, the athletes performed familiarization sessions once a week. During the familiarization sessions, each athlete performed 4 sets of 3 repetitions of the SQ under external compression (~100% AOP) against a load of 50% of their estimated 1-RM. The familiarization sessions were performed to restrict possible learning effects. One week before the main experiment, the 1-RM SQ test was performed. At the beginning of the warm-up, the subjects cycled on an ergometer for 5 min, followed by a general lower body warm-up. Next, the athletes performed 10, 6, 4, and 3 repetitions, starting at a load of 20kg and progressing to 60–80% of their estimated 1-RM. The first testing load was set to an estimated 90% 1-RM and was increased by 2.5–10 kg for each subsequent attempt until the athlete was unable to perform a proper lift with a correct technique. The 1-RM test result was determined within 5 sets of one repetition, with 5-min of rest between attempts [39]. All testing was performed with a constant tempo of movement [40,41]. The spotters and strength coaches were present throughout the procedure of 1-RM testing. The athletes started from an upright position, with the knees and hips fully extended, the stance approximately shoulder-width apart with both feet positioned flat on the floor in parallel or externally rotated to a maximum of 15° [42]. The bar rested across the back at the level of the acromion. Stance width and feet position were individually adjusted and carefully replicated on every lift. The bar was required to remain in contact with the back and shoulders at all times [42]. From this position, they were required to descend until making contact with the bench and then perform the concentric phase of the movement in an explosive manner [43,44]. The height of the bench was individually selected and allowed each participant to descend with the hips below the knee line according to the rules of the International Powerlifting Federation (IPF).

#### 2.3.2. Experimental Sessions

The athletes performed the SQ exercise under three different conditions: (1) control, without external compression (CONT); (2) intermittent external compression with pressure of 100% AOP (EC-100), and (3) intermittent external compression with pressure of 150% AOP (EC-150). During each testing protocol, the athlete performed 3 sets of 3 repetitions with a load 70% 1-RM of the SQ. For all trials, participants were required to use constant tempo of movement with a 2-s eccentric phase and maximal explosive intent in the concentric phase. Spotters were present to provide verbal encouragement and safety for the subjects. A linear position transducer system (Tendo Power Analyzer, Tendo Sport Machines, Trencin, Slovakia) was used for the evaluation of bar mechanics [45]. The measurements were made independently for each repetition and automatically converted into values of bar velocity. Peak power output (PP) and peak bar velocity (PV) were obtained from the highest results over the 3 repetitions. The mean power output (MP) and mean bar velocity (MV) were obtained as the mean of 3 repetitions performed in particular sets.

#### 2.3.3. External Compression

During an external compression session, subjects wore pressure cuffs at the most proximal region of each leg. For the experiment, we used Smart Cuffs (10 cm; Smart Tools Plus LLC, Strongsville, OH, USA). The individual values of pressure were administered according to previously published research [14]. Next, the pressure of external compression for the SQ was set to ~100% of full arterial occlusion pressure (173 ± 17 mmHg) or to 150% of full arterial occlusion pressure (256 ± 26 mmHg). The level of vascular restriction was controlled by a handheld Edan SD3 Doppler with an OLED screen and a 2 mHz probe made by Edan Instruments (Shenzen, China), placed on the posterior tibial artery. The external compression was applied immediately before the start of the set and released upon completion of the third repetition (Figure 1).

#### 2.3.4. Statistical Analysis

All statistical analyses were performed using Statistica 9.1. Results are presented as means with standard deviations. The Shapiro–Wilk, Levene, and Mauchly’s tests were used to verify the normality, homogeneity, and sphericity of the sample data variances, respectively. Differences between the CONT, EC-100, and EC-150 conditions were examined using repeated measures two-way (3 × 3; conditions × set) ANOVA. The statistical significance was set at *p* < 0.05. In the event of a significant main effect, post hoc comparisons were conducted using the Tukey’s test. Percent changes and 95% confidence intervals were also calculated. Effect sizes (Cohen’s *d*) were reported where appropriate. Parametric effect sizes were defined as large (*d* > 0.8), moderate (*d* between 0.8 and 0.5), small (*d* between 0.49 and 0.20), and trivial (*d* < 0.2) [46]. 

## 3. Results

The two-way repeated measures ANOVA indicated statistically significant main conditions effect for PP (*p* = 0.02), MP (*p* = 0.02), PV (*p* = 0.01), and MV (*p* = 0.01). The post hoc analysis for the main effect of conditions showed a statistically significant increase in PP (*p* = 0.03), PV (*p* = 0.02), MP (*p* = 0.04), and MV (*p* = 0.03), for EC-150, when compared to CONT (Table 1). Further, the post hoc analysis for the main effect of conditions showed a statistically significant increase in PP (*p* = 0.04), PV (*p* = 0.03), MP (*p* = 0.02), and MV (*p* = 0.01), for EC-150, when compared to EC-100 (Table 2). There were no significant differences in PP, PV, MP, and MV, between EC-100 and CONT. The analysis of the effect size are shown in Table 2. 

## 4. Discussion

The main finding of the present study was that external muscle compression significantly increases PP, MP, PV, and MV, during the SQ exercise. However, only EC-150 conditions showed a statistically significant increase in all considered variables, compared with EC-100, and with CONT, while such changes were not observed in EC-100 vs. CONT. Therefore, the results of this study indicate that only extremely high pressure of external compression (150% AOP) was effective in eliciting an increase in power output and bar velocity during the SQ. 

Currently, only one previous study has analyzed the acute impact of external compression on power output and bar velocity during a high loaded resistance exercise [14]. Wilk et al. [14] showed a significant increase in power output and bar velocity during the bench press exercise under external compression and at load 70% 1-RM. Despite using the same device “Smart Cuffs” with the same width 10cm as in the present [14], the results are contradictory. Wilk et al. [14] showed a significant increase in power output and bar velocity under external compression with pressure of 90% AOP; however, the presented study did not show such an increase under pressure of 100% AOP. However, in the study of Wilk et al. [14] the external compression was used in the upper limbs, while in this experiment the compression was applied to the lower limbs. As suggested by Crenshaw et al. [47] and Loenneke et al. [5], the absolute value of pressure used for external compression depends largely on the circumference of the limb to which the compression is applied. Therefore, the lower limbs, due to their larger circumference, to produce a similar effect require a higher absolute pressure than the upper limbs [7]. The increase in compression pressure by 10% AOP for the lower limbs, compared to the study of Wilk et al. [14] (upper limbs), was not sufficient to induce an increase in power output and bar velocity. However, the additional pressure of compression increased to 150% AOP was sufficient to induce an increase in performance during the SQ. Therefore, the result of the present study as well as the study by Wilk et al. [14] confirm that the potential effect of external compression on acute changes in power variables may be related to limb circumference and the pressure of the cuff [3,32,48,49]. Furthermore, the effect of the surface on which the compression is applied may depend not only on cuff width but also on the length of the occluded limb. The lower limb may require a wider cuff, to induce a similar acute performance enhancement, as observed for the upper limbs due to the proportions of the circumference and length of the limb to the width of the cuff. It seems that this aspect requires further research. However, the presented results, as well as those of Wilk et al. [32], indicate that the ratio of cuff width to the length of limb occluded can be compensated for by an increase in pressure of external compression. 

Furthermore, Wilk et al. [14] suggested that not only the physiological factor but also the mechanical energy generated by the cuff is a potential factor affecting the acute increase in strength and power performance under external compression. A significant increase in PP, PV, MP, and MV, for EC-150, when compared to EC-100 observed in the presented study occurred, although full blood flow restriction was applied in both conditions. Therefore, the physiological level of metabolic stress related to full BFR was similar for both conditions, which suggests that other factors had an impact on the increase in SQ performance for EC-150 conditions when compared to EC-100. Similar results (but for upper limbs) were observed in the study of Wilk et al. [32], who showed an increase in 1-RM test results, as well as an increase in the number of performed repetitions and time under tension during the bench press exercise under external compression with 150% AOP, compared to 100% AOP. Therefore, the results of this study as well as the previous study of Wilk et al. [32] showed that an additional increase in external compression pressure above complete blood flow restriction may additionally enhance performance. As previously suggested by Wilk et al. [14,32] and Rawska et al. [29], the mechanical energy generated by the cuff, strictly external, may have a potential impact on performance, which may partially explain the obtained results. A cuff used for external compression is a passive element, but during the movement (especially in the eccentric phase) the strain of the material of which the cuff is made may produce additional elastic energy [14], which can consist of the main factor affecting the increase in performance for EC-150, compared to EC-100, as well as compared to the CONT. Such an effect may be similar to that which occurs when using special compression garments, as during a squat in powerlifting, which supports the lifter during the eccentric phase of the movement, and by the accumulation of mechanical energy resulting from the tension of the material providing the “rebound” effect during the concentric phase of movement [50,51]. During sporting activities, the use of compressive garments can increase the cardiac capacity and the circulatory flow of the whole body, causing a vasodilatation and an increase in the venous flow [52]. The fact that compression therapy could have positive effects on blood flow and venous return in patients was sufficient for its introduction in the sports field, where small improvements in the body of the athlete can produce large increases in performance. Compression garments have been used for different parts of the body, such as foot-knee, foot-hip, thigh, and upper limbs, or they can cover larger body areas [53].

Even though the results of this study may be important for sports performance, the studies also have limitations that should be mentioned. It is often speculated in the literature that thigh circumference or composition of the limbs may restrict flow differently between individuals, which may account for some of the variability in the response to external compression [5,48,54] that was also observed in the presented study. Therefore, some individuals do not benefit from exercise under mechanical compression, which indicates the need for individualization in the use of external compression as a tool to increase performance. Furthermore, the use of extremely high-pressure mechanical compression caused high discomfort as well as skin injuries [32]. Additionally, even if high external compression causes an acute increase in strength and power performance, this effect does not necessarily translate into chronic changes. It should also be remembered that excessive frequency of external compression may have negative consequences. Frequent exercise under external compression applied in same muscle area can impair muscle structure directly in the region under the cuff, which is a risk not only for sports performance but also for the health of the athlete [55,56]. Although the results showed that external compression may have a significant impact on power output and bar velocity during the SQ, the causes of these changes could not be determined and explained due to the lack of physiological and biomechanical evaluations, which could explain this phenomenon in more detail. 

### Practical Implications

The results of the present study indicate that short-term and high-pressure external compression increases power output and bar velocity during the SQ exercise with a load of 70% 1-RM. Therefore, the external muscle compression can be an important tool used for acute performance enhancement. The presented experimental protocol can be easily reproduced in resistance training sessions both for lower limb as well as upper limb. However, for lower limb only extremely high compression was effective in eliciting acute power output and bar velocity improvements. Furthermore, an acute increase in the level of performance under external muscle compression does not provide certainty about long-term adaptive changes, and these results cannot translate to other values of external load. Additionally, due to the possibility of impaired muscle structure directly in the region under the cuff, the external compression should not be used in every set of a particular exercise or in every training unit. It seems that a safe and optimal solution utilizing the positive impact of external compression on performance constitutes both: combined sets with and without external compression as well as changing the area of cuff application on the limb. 

## 5. Conclusions

The presented results confirm advantages of combining extremely high pressure of external compression with high external loads (70% 1-RM) during resistance exercise is an effective strategy for increasing power output and bar velocity during several sets of SQ exercise. Therefore, such interventions could be a useful tool to enhance training programs and would allow additional opportunities to modify resistance training routines, breaking training monotony, breaking through sticking points, and increasing power output, especially in competitive athletes. However, there is still a need for further research, especially with the use of high-pressure external compression during specific sport activities, such as athletics throws, jumps, or sprints.

## Figures and Tables

**Figure 1 ijerph-17-04674-f001:**
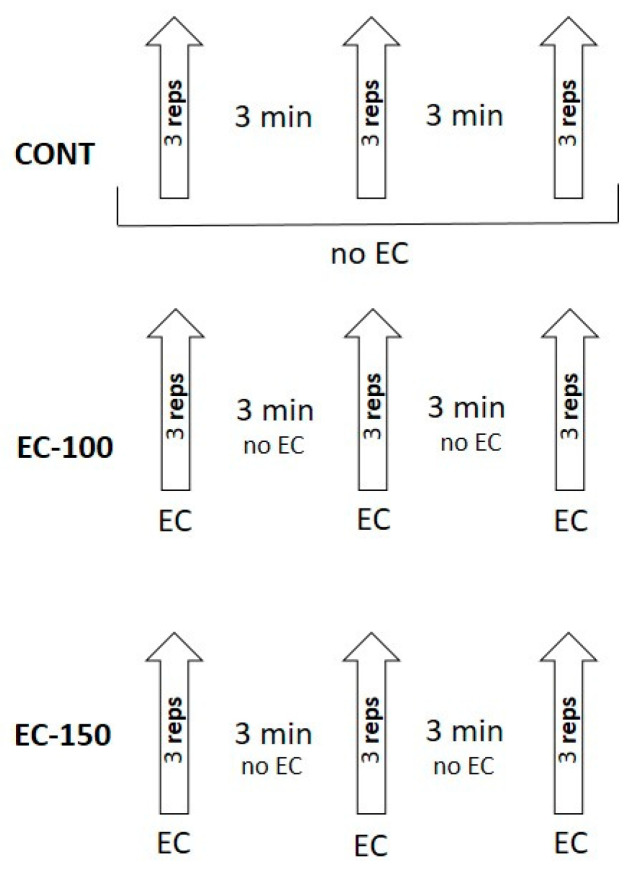
The experimental design. Control, without external compression (CONT); external compression with pressure of 100% AOP (EC-100); external compression with pressure of 150% AOP (EC-150).

**Table 1 ijerph-17-04674-t001:** Main statistical differences between exercise conditions.

Condition	CONT vs. EC-100	CONT vs. EC-150	EC-100 vs. EC-150
Peak Power Output [W]	0.97	0.03 *	0.04 *
Peak Bar Velocity [m/s]	0.97	0.02 *	0.03 *
Mean Power Output [W]	0.94	0.04 *	0.02 *
Mean Bar Velocity [m/s]	0.94	0.03 *	0.01 *

* Statistically significant difference *p* < 0.05. Control, without external compression (CONT); external compression with pressure of 100% AOP (EC-100); external compression with pressure of 150% AOP (EC-150).

**Table 2 ijerph-17-04674-t002:** Summary of performance data under the three employed exercise conditions.

Squat	CONT (95% CI)	EC-100 (95% CI)	EC-150 (95% CI)	EFFECE SIZE		
				CONT vs. EC-100	CONT vs. EC-150	CONT vs. EC-150
**Peak Power Output (W)**						
Set 1	2080 ± 443 (1763 to 2397)	2060 ± 336 (1819 to 2300)	2170 ± 412 (1875 to 2465)	0.05	0.21	0.29
Set 2	2134 ± 428 (1828 to 2441)	2129 ± 309 (1908 to 2350)	2249 ± 545 (1859 to 2638)	0.01	0.23	0.27
Set 3	1971 ± 411 (1677 to 2265)	2038 ± 359 (1781 to 2294)	2252 ± 484 (1906 to 2598)	0.17	0.63	0.50
**Peak Bar Velocity (m/s)**						
Set 1	1.49 ± 0.16 (1.38 to 1.60)	1.48 ± 0.14 (1.38 to 1.58)	1.56 ± 0.08 (1.50 to 1.62)	0.07	0.55	0.70
Set 2	1.52 ± 0.14 (1.42 to 1.62)	1.53 ± 0.11 (1.45 to 1.60)	1.56 ± 0.10 (1.49 to 1.64)	0.08	0.33	0.29
Set 3	1.45 ± 0.15 (1.35 to 1.56)	1.48 ± 0.13 (1.38 to 1.57)	1.57 ± 0.12 (1.48 to 1.66)	0.21	0.88	0.65
**Mean Power Output (W)**						
Set 1	811 ± 248 (634 to 989)	785 ± 193 (646 to 923)	865 ± 266 (674 to 1055)	0.12	0.21	0.34
Set 2	816 ± 248 (638 to 993)	808 ± 196 (668 to 948)	879 ± 275 (682 to 1076)	0.04	0.24	0.30
Set 3	794 ± 221 (636 to 952)	805 ± 219 (648 to 961)	867 ± 233 (700 to 1034)	0.05	0.32	0.27
**Mean Bar Velocity (m/s)**						
Set 1	0.77 ± 0.12 (0.69 to 0.86)	0.75 ± 0.07 (0.70 to 0.80)	0.82 ± 0.09 (0.75 to 0.88)	0.20	0.47	0.87
Set 2	0.77 ± 0.10 (0.70 to 0.85)	0.77 ± 0.05 (0.73 to 0.81)	0.83 ± 0.10 (0.76 to 0.90)	0	0.60	0.76
Set 3	0.76 ± 0.10 (0.68 to 0.83)	0.77 ± 0.11 (0.69 to 0.84)	0.82 ± 0.08 (0.77 to 0.88)	0.10	0.66	0.55

All data are presented as mean ± standard deviation. Control, without external compression (CONT); external compression with pressure of 100% AOP (EC-100); external compression with pressure of 150% AOP (EC-150); CI: confidence interval.

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
