# Peer review of "The Acute Impact of External Compression on Back Squat Performance in Competitive Athletes"

_ijerph, 2020, doi:10.3390/ijerph17134674_

Round 1
Reviewer 1 Report
The following study sought to describe the effects of compression/BFR on power and velocity of barbell movement during back squat in judo athletes. Overall, I like the focus of this study. The manuscript is written well, and findings have important implications in using BFR to optimize training techniques. Below are my specific comments.:
This should be considered a “brief report”. Not because the results are not impactful or interesting, simply because much of the data/tables shown are not needed and could be condensed to help with readability.
Line 25: change to “during back squat exercise”
Line 25: Decimal places should be consistent throughout. In this case, I would recommend to the tenths place as going past this goes beyond what was likely discernible during measurement.
Line 29: Authors should number the conditions here.
Line 68: Remove “the study by”
Line 72: remove “is” and add “may be”
Line 97: purpose statement should be given here. Additionally, the hypothesis should be more specific. Which aspects of performance? Peak and mean values or just one or the other? While mentioned in the abstract, no mention of power output and barbell velocity measures for this study have been mentioned in the intro.
Overall, the authors have done an excellent job with explaining the rationale and need for their study in the introduction.
Line 101: This paragraph should have a subheading which should read “2.1 Study Design”. Numbers of subheadings should be changed thereafter.
Line 101: Conditions were counterbalanced?
Line 108: “maximal tempo” seems odd. Furthermore, the term “constant tempo” for the eccentric phase is used later which complicates things. I would replace it with “maximal explosive intent”.
Line 111: change to “Male judo athletes (n=10)”. If not all males, sex breakdown should be noted.
Line 113: Authors should add relative strength here (1RM kg/ BM kg)
Line 147: A lot of this is repeated information and should be removed.
Line 157: The description of the peak and mean values is a little confusing. For peak power, does that denote peak power during that one repetition or simply the highest mean power output over the 3 repetitions? The former is typically what I am accustomed to measuring. Thus, you should get a peak value and mean value for each repetition completed. So, averaging all three values together for both peak and mean values is more appropriate.
Line 157: What is the relevance of reporting peak values in addition to the mean values? To me, peak values are more appropriate for vertical jump or other movements which may have a flight phase to the movement. Most evidence suggests mean values are more indicative of performance during resistance exercise and have greater implications for training optimization.
Line 192: Some of the tables are not needed. There is no need for a table on main effects because that is already reported in the results section. Table 2 and 3 could be combined. As shown, these tables appear to presented just to “fill space”. This does not detract from the scientific soundness of the study, merely just should be presented for what it is. Since the sets are non-fatiguing sets, showing set to set does not add anything to answer the research question and should be removed. This is in part why I believe this manuscript should be considered a “brief report”.
Line 233: Change “is” to “may be”. Absolute language like this should be changed throughout unless directly assessed though the study design. For example, the current study did not elucidate how limb circumference relates to efficacy of BFR and performance. Thus, while the rationale is strong for this mechanism, it was not directly measured and should be approached with more conservative language.
Line 259: So, when authors say “mechanical energy”, is this strictly extrinsic? Are authors suggesting that compression on the muscle stretches sarcomeres allowing for greater cross bridge formation (i.e. intrinsic) or it is simply an extrinsic force aiding in propulsion?
Line 263: Why is this?
Author Response
Reviewer 1
Thank you for your helpful comments and suggestions. The authors are also grateful for the Reviewer taking the time to evaluate this manuscript. We hope that the revised manuscript satisfactorily addresses all the raised issues. The responses to the Reviewer’s comments are below.
Please note that the line numbers in our replies refer to those of the present submission and that all changes in the manuscript are marked with yellow highlighting
The following study sought to describe the effects of compression/BFR on power and velocity of barbell movement during back squat in judo athletes. Overall, I like the focus of this study. The manuscript is written well, and findings have important implications in using BFR to optimize training techniques. Below are my specific comments.:
This should be considered a “brief report”. Not because the results are not impactful or interesting, simply because much of the data/tables shown are not needed and could be condensed to help with readability.
Reply - We agree with the opinion, however, we wanted to present the widest possible results, which allows us to increase the transparency of the obtained results.
Line 25: change to “during back squat exercise”
Reply – the change has been made
Line 25: Decimal places should be consistent throughout. In this case, I would recommend to the tenths place as going past this goes beyond what was likely discernible during measurement.
Reply – the change has been made
Line 29: Authors should number the conditions here.
Reply – the change has been made
Line 68: Remove “the study by”
Reply – the change has been made
Line 72: remove “is” and add “may be”
Reply – the change has been made
Line 97: purpose statement should be given here. Additionally, the hypothesis should be more specific. Which aspects of performance? Peak and mean values or just one or the other? While mentioned in the abstract, no mention of power output and barbell velocity measures for this study have been mentioned in the intro.
Reply – the specific information to hypothesis was added L 98
Overall, the authors have done an excellent job with explaining the rationale and need for their study in the introduction.
Line 101: This paragraph should have a subheading which should read “2.1 Study Design”. Numbers of subheadings should be changed thereafter.
Reply – the change has been made
Line 101: Conditions were counterbalanced?
Reply – yes, we added such information
Line 108: “maximal tempo” seems odd. Furthermore, the term “constant tempo” for the eccentric phase is used later which complicates things. I would replace it with “maximal explosive intent”.
Reply – the change has been made L 152
Line 111: change to “Male judo athletes (n=10)”. If not all males, sex breakdown should be noted.
Reply – the change has been made
Line 113: Authors should add relative strength here (1RM kg/ BM kg)
Reply – the change has been made
Line 147: A lot of this is repeated information and should be removed.
Reply – the change has been made
Line 157: The description of the peak and mean values is a little confusing. For peak power, does that denote peak power during that one repetition or simply the highest mean power output over the 3 repetitions? The former is typically what I am accustomed to measuring. Thus, you should get a peak value and mean value for each repetition completed. So, averaging all three values together for both peak and mean values is more appropriate.
Reply – the change has been made. Peak power output (PP) and peak bar velocity (PV) was obtained from the highest results over the 3 repetitions. The mean power output (MP) and mean bar velocity (MV) was obtained as the mean of three repetitions performed in particular sets. L 155- 159
Line 157: What is the relevance of reporting peak values in addition to the mean values? To me, peak values are more appropriate for vertical jump or other movements which may have a flight phase to the movement. Most evidence suggests mean values are more indicative of performance during resistance exercise and have greater implications for training optimization.
Reply – we decided to report peak values according to previous research related to the analysis of acute power output and velocity changes. 10.1519/JSC.0000000000003649
Line 192: Some of the tables are not needed. There is no need for a table on main effects because that is already reported in the results section. Table 2 and 3 could be combined. As shown, these tables appear to presented just to “fill space”. This does not detract from the scientific soundness of the study, merely just should be presented for what it is. Since the sets are non-fatiguing sets, showing set to set does not add anything to answer the research question and should be removed. This is in part why I believe this manuscript should be considered a “brief report”.
Reply - according to suggestion we delete table 1. However, for better clarity of results, we decided to keep tables 2 and 3 remaining.
Line 233: Change “is” to “may be”. Absolute language like this should be changed throughout unless directly assessed though the study design. For example, the current study did not elucidate how limb circumference relates to efficacy of BFR and performance. Thus, while the rationale is strong for this mechanism, it was not directly measured and should be approached with more conservative language.
Reply – the change has been made
Line 259: So, when authors say “mechanical energy”, is this strictly extrinsic? Are authors suggesting that compression on the muscle stretches sarcomeres allowing for greater cross bridge formation (i.e. intrinsic) or it is simply an extrinsic force aiding in propulsion?
Reply – yes, the mechanical energy is strictly external. We added such information. L 244
Line 263: Why is this?
Reply – this sentence was deleted.
Reviewer 2 Report
I would like to suggest some points about the manuscript.
- Try to make a experimental design figure.
- Please, check the statistical results of table 3, since the main significant differences were found between Cont vs EC-100 and not vs EC-150, which is reported in the conclusion.
- Since the results are quite interesting, and the experimental protocol could be easily replicated, the authors should highlight the practical applications of the study.
Are the authors able to justify any inconvenience of reducing movement range due to a high pressure cuff applied to the thigh? On the other hand, could the effects be caused by the pressure cuff of 150mm mechanical? And not due to blood flow occlusion effects? Also, the both pressures applied were able to promote blood flow occlusion? Yes! So, how to explain the increases in performance due to blood flow changes? The authors could try to compare the use of compression garments in the discussion section.
Author Response
Thank you for your helpful comments and suggestions. The authors are also grateful for the Reviewer taking the time to evaluate this manuscript. We hope that the revised manuscript satisfactorily addresses all the raised issues. The responses to the Reviewer’s comments are below.
Please note that the line numbers in our replies refer to those of the present submission and that all changes in the manuscript are marked with yellow highlighting
I would like to suggest some points about the manuscript.
Try to make a experimental design figure.
Reply – the design figure was added
Please, check the statistical results of table 3, since the main significant differences were found between Cont vs EC-100 and not vs EC-150, which is reported in the conclusion.
Reply – there was main effect but it was not significant interaction. In the absence of statistical interaction according to the statistics section, no post hoc test was performed. Therefore, in discussion the main effect was described, not effect for individual sets.
Since the results are quite interesting, and the experimental protocol could be easily replicated, the authors should highlight the practical applications of the study.
Reply – we added new information to practical applications. L 281-282.
Are the authors able to justify any inconvenience of reducing movement range due to a high pressure cuff applied to the thigh? On the other hand, could the effects be caused by the pressure cuff of 150mm mechanical? And not due to blood flow occlusion effects? Also, the both pressures applied were able to promote blood flow occlusion? Yes! So, how to explain the increases in performance due to blood flow changes? The authors could try to compare the use of compression garments in the discussion section.
Reply – the were no statistical differences in ROM between control and BFR conditions.
Both pressures applied caused shutdown blood flow, so the increase in performance was related to the mechanical energy generate be the cuff, comparable to this what occurs during the use compression garments. We added a new statement. L 253-259.
Round 2
Reviewer 1 Report
The authors have done a nice job here. They were very respectful and clear with edits. Thank you for allowing me to review your work.